# May a Pair of 'Eyes' Be Optimal for Vehicles Too?

**Ernst D. Dickmanns**

Independent Researcher, 85649 Brunnthal, Germany; edd@dyna-vision.de

**Abstract:** Following a very brief look at the human vision system, an extended summary of our own elemental steps towards future vision systems for ground vehicles is given, leading to the proposal made in the main part. The question is raised of why the predominant solution in biological vision systems, namely pairs of eyes (very often multi-focal and gaze-controllable), has not been found in technical systems up to now, though it may be a useful or even optimal solution for vehicles too. Two potential candidates with perception capabilities closer to the human sense of vision are discussed in some detail: one with all cameras mounted in a fixed way onto the body of the vehicle, and one with a multi-focal gaze-controllable set of cameras. Such compact systems are considered advantageous for many types of vehicles if a human level of performance in dynamic real-time vision and detailed scene understanding is the goal. Increasingly general realizations of these types of vision systems may take all of the 21st century to be developed. The big challenge for such systems with the capability of learning while seeing will be more on the software side than on the hardware side required for sensing and computing.

**Keywords:** automotive vehicles; autonomous driving; vision systems

## 1. Introduction

The acceptance of technical vision systems by the human driver depends much on the differences in performance the overall system shows in comparison to his own capabilities. Therefore, in Section 1.1, a brief survey is given on some characteristic parameters of the human vision system. Section 1.2 then presents a review of the elemental steps towards future vision systems of ground vehicles, leading to the proposal made in Section 4. In Section 2, a brief survey of the reasons behind the different developments in biology and technology is laid out. Section 3 contains some conclusions for further developments. An outlook on potential future paths of development in automotive vision systems is given in Sections 4 and 5. Based on observations made, both in many biological species and in technical developments, two (of many possible) designs are looked at in some more detail. Neural net approaches are not taken into account for various reasons. The author is convinced that—as the 20th century has seen a wide spread of designs for ground vehicles—the 21st century will see the development of a wide variety of technical vision systems for vehicles, including deep neural nets. These developments are not a matter of years but of decades.

### 1.1. Characteristics of the Human Eye and Brain

The larger part of the brain of highly developed vertebrates (including ourselves) is devoted to the processing of visual data for scene understanding, behavior decision and motion control [1–4]. Many biologists consider vision as one of the driving factors for developing intelligence, evolved by individuals with complex brains over millions of years [5]. Many different types and numbers of eyes are found per individual in a wide range of classes of animals. It is hard to believe that only chance is behind the fact that most animals with high-performance locomotion capabilities on land, in water and in the air have pairs of (mostly gaze-controllable) eyes with radial areas of different resolutions in

the images they provide. This modality of vision must have advantages (at least for carbon-based systems) that have evolved over many generations. Figure 1 shows some characteristics of one of these solutions, the human eye (from p. 5 in [6]). The total horizontal field of view (FoV) of a single eye is about 145°; the image resolution decreases from the center to the periphery. The central area, dubbed 'fovea', a region of slightly elliptical size, has a visual range of about 1.5° vertically to about 2° horizontally. There are two different types of sensor elements present in the eye: Color sensitive 'cones' and only-intensity-sensitive 'rods'. The fovea contains exclusively cones, about 7 million (blue curve in Figure 1) with a high density of about 150,000 receptors per mm$^2$; this allows a 'Vernier acuity' in edge detection in the range of 0.06 to 0.25 mrad (individual variations). For angles radially away from the fovea > 15° (in the peripheral regions of the retina), the density of cones is about 1/30 of the peak value; that means that the distance between cones here is five to six times that between rods.

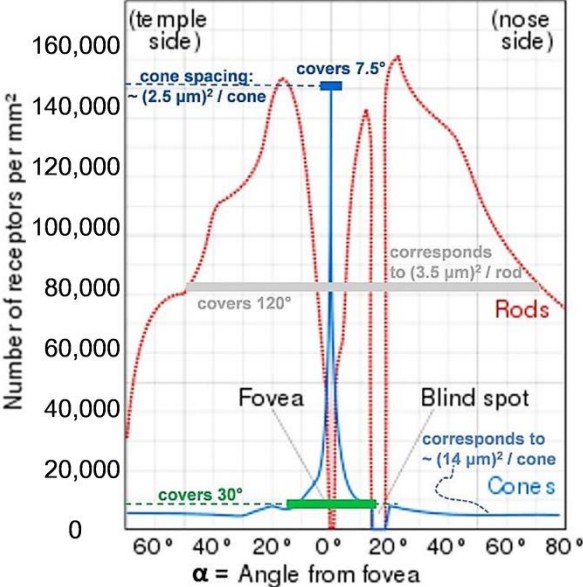

**Figure 1.** Distribution of rods and cones along a line passing through the fovea and the blind spot of a human eye (p. 5 in [6]).

The fovea is employed for accurate vision in the direction toward where it is pointed by the rotation of the eye. It comprises less than 1% of the retinal size, but uses over 50% of the visual cortex in the brain [7]. The ratio of ganglion cells to photoreceptors in the eye is about 2.5; almost every ganglion cell in the eye receives data from a single cone, and each cone feeds into one to three ganglion cells. Image preprocessing is performed in the eye so that the number of nerves feeding the primary visual cortex in the back side of the human brain is reduced by two orders of magnitude compared to the number of photoreceptors in the eye; edge orientations are coded directly in the eye to achieve this reduction [8]. The eye does not take a photographic image of the environment, but rather produces in parallel up to 100 extremely compressed full images of low precision for the peripheral FoV; in addition, three to four small areas with high resolution are generated by foveal perception [9]. These data are compared with existing images in the actual imagination process, and both together are transformed into the actual perception of the environment.

The human eye has three angular degrees of freedom: Two for gaze directions of the line of sight and one for a rotation around it. The latter, called roll angle, will be neglected here. The downward looking angle (infraduction or depression) may go to −60°, the upward looking angle (supraduction or elevation) to +45°. The capability of the horizontal rotation may go on average up to 50° both towards the nose and away from it; however, in general the angles hardly exceed ±20°. These changes in the viewing direction are supported by rotations of the head. Fast eye movements may go up to 600°/s in saccades; during the tracking of objects (fixation), the maximal rotational speed is about

100°/s. Saccades are usually finished after 50 milliseconds (ms) [6]. Fixations and saccades make up the largest part of cognitive eye movements. Thus, intentional vision is a controlled cognitive activity by the subject.

The red curves in Figure 1 show the density of the (b/w) elements that measure intensity-values only. There are no rods in the fovea; their density rises rapidly in the first 15° radially away from the fovea to similar values to those that cones have in the fovea; keep in mind that Figure 1 is a cut through the fovea and the 'blind spot', which is the circular area where visual nerves pass through the retina. For other cuts through the retina, the red curves at the left- and right-hand sides of the figure are connected by some kind of inverted parabola. At a radial distance of around 15° away from the fovea, rod spacing is as high as cone spacing in the fovea (about 2 to 2.5 μm between elements). In a FoV of 120° (50° temple side to 70° nose side, see the inserted gray bar) the number of receptors per mm$^2$ is above 80,000, which is about half that of the fovea (and of the imaginary center); this value corresponds to a spacing of about 3.5 μm between the sensor elements. These numbers are, of course, specific to the carbon-based wetware of all biological systems. The basic distinctions to silicon-based hardware will be addressed in more detail in Section 2.

### 1.2. The Development of the Actual State of Technical Real-Time Vision Systems

In Japan, S. Tsugawa in the late 1970s investigated the lateral guidance of a road vehicle by using analog data processing in image analysis for stereo vision in vertical planes; single rows of the images of two cameras—mounted above each other (see Figure 2) and rotated by 90°—were evaluated in order to detect the guide rails at the side of the road [10]. All electronic devices that were needed were on board. Short distances were driven at a low speed. In the early 1980s, there was a gap with respect to vision-guided road vehicles in Japan. The activities were resumed when new results from the USA and Germany became public in the second half of the decade.

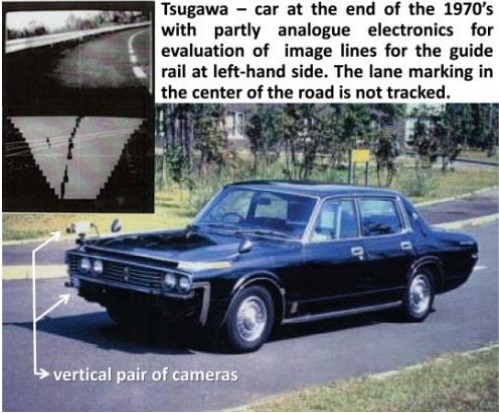

**Figure 2.** The first vision-guided road vehicle in the late 1970s, Japan [10].

### 1.2.1. Vision Systems on Digital Microprocessors with All Equipment on Board

The development of real-time vision systems based on digital microprocessors (μP), with all the equipment needed on board the vehicle, started in the USA and in Germany, independently of each other. In both countries, the vehicles chosen at the beginning were vans with weights from 5 to 8 tons. In the USA, the huge DARPA project "On Strategic Computing" started in 1982 [11]; the 'Autonomous Land Vehicle' (ALV) was one of three fields of application. Real-time computer vision was an essential element, since video sensors provide a much higher signal density and better capabilities for object recognition in comparison to other types of sensors. Laser-Range-Finders (LRF) have been incorporated right from the beginning for the easy determination of range to points in the environment and to objects. Carnegie Mellon University (CMU) had its own vehicle, NavLab, a van specially equipped for slow driving. At the core of the project were massively parallel computer systems for data analysis

and dynamic scene understanding to be developed over the next decade or two. In the meanwhile, several universities and research institutes were funded in order to develop software for interpreting and understanding image sequences on computers that were actually available. The typical cycle times for a single perception step were in the range of seconds to minutes.

All these groups coming from Computer Science (CS) and Artificial Intelligence (AI) started by investigating rather complex vision tasks at a much slowed-down rate on available computing hardware, while at the same time massively parallel computer systems were investigated for analyzing sequences of digital images [11]. Temporal aspects were considered separately in a second step. Differences in the interpretation results of consecutive images had to be interpreted either as movements of objects in the mapped 3-D scene, or as changes due to ego-motion (or both in conjunction). A completely different approach has been chosen by the author, coming from systems dynamics and control engineering, including optimal control. A detailed survey of the history of vision for ground vehicles may be found in [12].

**The 4-D approach to dynamic vision:** In Germany, the author had experience in working with horses in agriculture when in the 1950s a tractor replaced horses. He noted that the advantage of the tractor having multiple horsepower was sometimes offset by the disadvantage of having to pay attention to vehicle guidance all the time. Horses found their way home by themselves and even kept the learned distance to the road boundary quite well while moving. Since tractors had no sense of vision, the human driver had to be alert all the time. It became clear to him that developing a sense of vision for vehicles should be one important task for the future. After studies of mechanical and control engineering and a few years of working in flight mechanics, in 1975 he was appointed professor for control engineering at the newly founded University of the Federal Armed Forces of Germany in Neubiberg near Munich (UniBwM). He realized that the remaining 26 professional years corresponded to an increase in computing power by five to six orders of magnitude; this number was based on the observation over the recent past that computing power increased by a factor of ten every four to five years. Trusting in further miniaturization and in the observed temporal increase in computing power, a real-time evaluation of video sequences at around 10 Hz should be achievable with a modest number of parallel μP within about 20 years.

At that time, he did not know about the activities in the USA and in Japan. He used the initial funding of UniBwM to start his long-term development effort by building 'Hardware-In-the-Loop' (HIL) simulation facilities for machine vision systems similar to those on the market for training pilots [13]. When studying the visual guidance of aircraft landing approaches in the 1980s, it turned out that without the feedback of inertial sensor signals from the gaze-platform to the image interpretation process, the approach that will be discussed below would not work under stronger winds and gusts. With this feedback, it has been shown to work satisfactorily [14]. All the experiences with real-world systems has led the UniBwM group to include gaze control in the software development for dynamic vision from the beginning, whenever possible.

The group selected sufficiently simple tasks for the beginning, but wanted to do real-time interpretation right from the start by using standard tools for the state estimation of dynamical systems. Initially, the 'Luenberger observer' [15] had been chosen in connection with 3-D shape models for the objects perceived, and with scene models including the differential equations for the motion components of the object observed in 3-D space over time. Since models for motion and for perspective projection are in general nonlinear, linearized approximations for short periods of time had to be selected [16–20]. Due to the fact that in practical life almost every process is subjected to unknown perturbations, the transition to (Extended) Kalman Filtering (EKF) was soon made [17,19]. A lower limit of interpreting at least ten frames per second was introduced as a side constraint. In connection with the feedback of prediction errors as a basic correction step, this brought about a different kind of thinking for the overall approach. Note that this method leads to a direct representation of the scene observed in 3-D space and time, i.e., in 4-D; it might thus be seen as a first step towards consciousness.

By exploiting the feedback of the prediction errors, only information from the last image of the sequence needed to be stored; the results of all the previous evaluations were stored in the best estimates for the shape- and state-values of the hypothesized (dynamical) objects. In terms of understanding road scenes, this approach had initially been tested since 1980 in a special Hardware-In-the Loop (HIL) simulation [13]. Using road representation in terms of differential geometry made the 4-D approach particularly efficient [16]. The positive results for the autonomous visual guidance of road vehicles led in 1984 to its implementation in a 5-ton van (see Figure 3a). It had to be specially equipped with about 3 m-high standard industrial racks for all the electronic equipment like communication devices and computers (Figure 3c). The sensors for vision were two cameras (Figure 3b) on a gaze control platform (seen at right in (Figure 3d) from the other side). The test-vehicle was dubbed "VaMoRs" (Versuchsfahrzeug für autonome Mobilität und Rechnersehen). The first results in real-world autonomous visual driving were achieved in 1986; in 1987, VaMoRs became the first autonomous vehicle capable of high-speed driving on roads when it achieved 60 mph (96 km/h, its maximum speed due to limited engine power) on a new stretch of the Autobahn not yet turned over to the public [20].

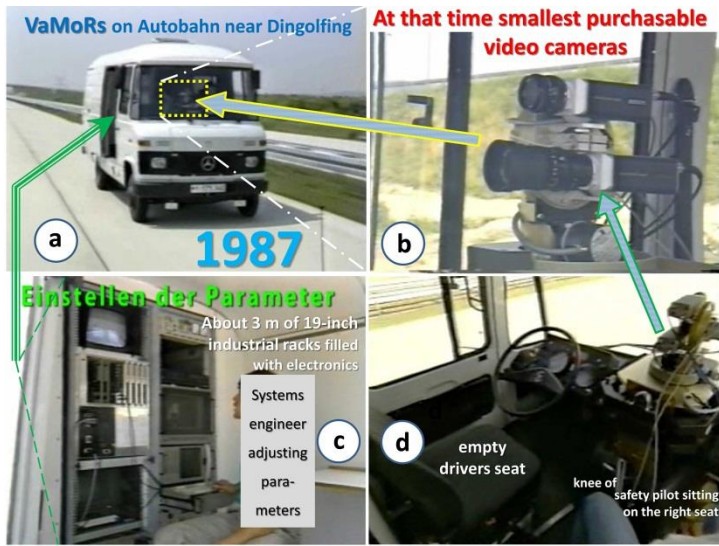

**Figure 3.** The first high-speed autonomous vehicle, VaMoRs, from 1986 to 2004 with three generations of vision systems [21] (sub-images from video 'VaMoRs Autobahn Dingolfing 1987').

These results motivated the company Daimler-Benz AG (DBAG) to join with UniBwM in order to develop the sense of vision for the autonomous guidance of road vehicles. This task became one of the activities in the common European framework 'EUREKA' for long-term research and developments. The Europe-wide 'PROgraMme for a European Traffic of Highest Efficiency and Unprecedented Safety' (PROMETHEUS) was modified by substituting 'machine vision' in exchange for lateral guidance of road vehicles based on electro-magnetic inductive fields generated by cables buried in the center of the lane. More than a dozen European automotive companies and about five times that number of research institutes and universities joined the sub-project, dubbed PRO-ART (derived from PROmetheus ART-ificial Intelligence). The program ran from 1987 until the end of 1994. The open exchange between the European participants led to the fact that all major automotive companies had their own vehicle(s) equipped with body-fixed cameras and capable of lane-recognition and -following.

When in the early 1990s the England-based 'Transputers' (µP with four parallel links with a transfer rate of up to 20 megabit/s to their neighbors) appeared on the market, it allowed for the reduction of vehicle size due to less storage space being required. Two high-end sedans Mercedes SEL-500 were equipped with a new central visual perception system based on transputers and the 4-D approach [22–26]: The DBAG-vehicle "Vision Information Technology Application" (VITA_2) and the

UniBwM-vehicle VaMoRs-PKW (in short VaMP, see Figure 4). As can be seen, the electronic devices needed for vision still occupied a relatively large volume.

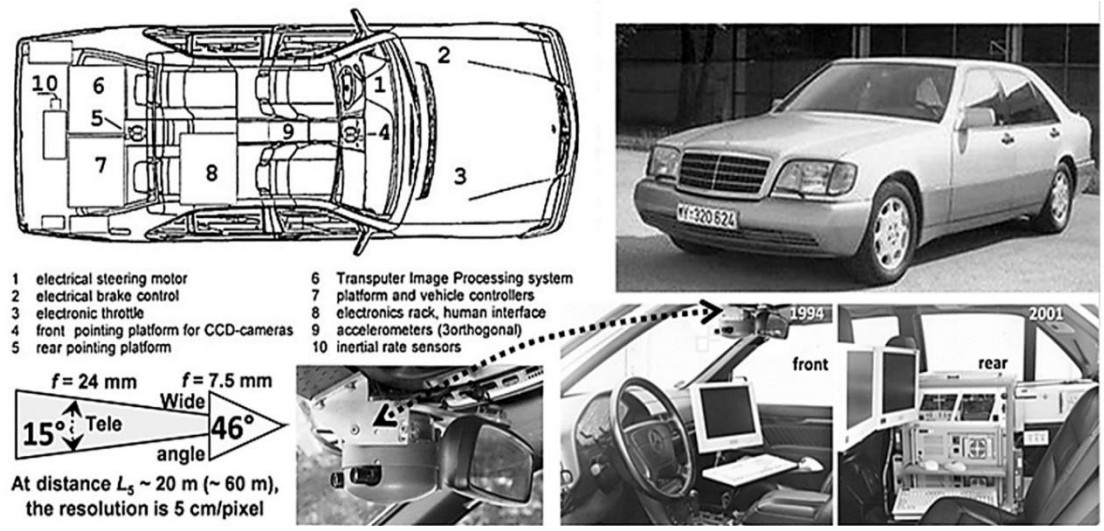

**Figure 4.** Demonstration vehicle VaMP of UniBwM (top right, a twin to the Daimler-VITA_2); top left: components for autonomous driving. Bottom right: A view into the passenger cabin with the first small bifocal "Vehicle eye" on a yaw-platform, detailed at bottom left.

All central cognitive components based on the two 'vehicle eyes' with new, small 'finger cameras' of about 1 cm in diameter mounted on yaw platforms of about 12 cm in diameter (see Figure 4, lower left) have been developed by UniBwM. Since highways are rather planar and smooth, and the vehicles were comfortable with only minor perturbations in pitch, the vertical degree of freedom was left off in order to achieve a simple and small design. The system is described in [23,24].

During the tests it became apparent that for a human-like perceptual performance in the long run several additional features would be necessary: 1. to extend the image resolution to higher values for an earlier detection and finer perception of details; 2. to have more cameras available for perceiving the nearby environment, and 3. to realize the capability of perceiving areal features like homogeneous regions in black-and-white or even in color and texture. Exploiting the advantages of the 4-D approach, stereo vision would only be necessary nearby, where the point at which an object touches the ground is not visible. Further away, environmental objects allow performing range estimation based on recognized known objects (like road width, size in the image, etc.).

To handle these challenges, DBAG added, for VITA_2, fourteen additional cameras distributed around the vehicle for collecting additional information on the immediate environment (see Figure 5). Their role is described in [27]; they were not used by the perception system of UniBwM since the computing power was missing. In the main part of this article, another arrangement of cameras assembled into an 'eye', probably on gaze-controllable platforms, at the top end of the structure holding the front windshield at both sides (A-frames) of a car will be discussed, which could perform the task of the safety driver in the mentioned Paris-demonstrations.

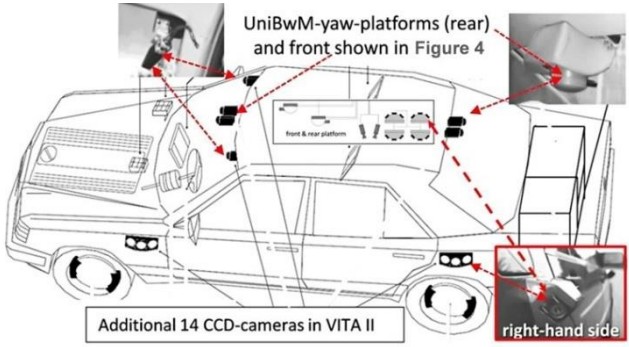

**Figure 5.** Camera arrangement in VITA_2 of Daimler 1994 to 1996. The rectangular insert in the roof-region shows the abstract scheme for positioning cameras (basic structure from [27]).

At the same conference in 1994 [28], a second-generation two-axis platform for VaMoRs was presented. Its dynamic performance was tested by tracking a triangular traffic sign at the side of the road when approaching it. This platform was capable of executing a 20°-saccade in less than 150 ms. A video from 1996 showing the characteristics, also in slow motion, may be viewed in [29]. The resulting graph of the experiment is shown in (Figure 6): Curve 1.1 (green, left) shows the change of the azimuth angle during the approach. Saccade-1 brings the sign into the FoV of the high-resolution camera (red curve left). It takes five evaluation cycles until the sign is tracked in the high-resolution image (green curve 1.2, bottom and center). This image is then stored and sent to the evaluation process; afterwards, the second saccade (red curve at right) brings the viewing direction back to normal again. Similar gaze maneuvers are needed for observing traffic lights; however, they have to be tracked over time since they are time-variable. As stated in [30], this task may be a major challenge when driving in cities and on different types of state roads; the positioning of traffic lights on both sides of the road (for turn-offs) and even high above the center of the road makes the task difficult. Fixation by a high-resolution camera, while other cameras with a wider FoV keep the overall traffic situation imaged in parallel, seems to be a better approach than body-fixed cameras.

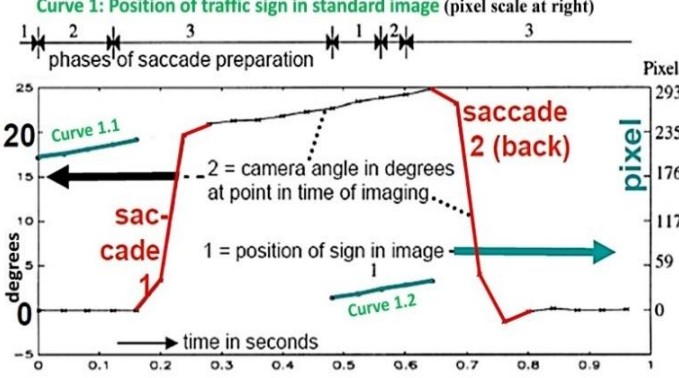

**Figure 6.** Detection and tracking of a traffic sign (curve 1.1); saccadic change of the gaze direction onto the sign (1), five image cycles for stable recognition, then reading the sign and storing the high-resolution image (curve 1.2); saccade 2 back to the original gaze direction (see [29]).

The different approaches were presented in new journals like 'Machine Vision and Applications' since 1988 [20]. On the IJCAI 1989 in Detroit, the 4-D approach and its applications were presented in an invited keynote address. Towards the end of the 1980s, research on autonomous ground vehicles in the USA was transferred to the Army Research Laboratory (ARL) in Aberdeen, Maryland. Its main research partner in autonomous driving was the National Institute of Standards and Technology (NIST)

nearby. ARL and NIST favored a common research project with UniBwM in Germany (reviewed below).

Proponents of the two approaches in the USA and Germany first met at a conference in 1986 [31] and at a high-level international symposium on robotics [32] in 1987. At this symposium, the new start in Japan in 1987 also became public. Afterwards, the development of the field of autonomous driving was promoted by frequent exchanges between the leading nations at international conferences and symposia. In 1996, ARL, along with NIST, proposed a joint US–American/German project for merging the best components developed so far on both sides.

**The joint US–American/German project AutoNav:** In the framework of an existing 'Memorandum of Understanding', a joint project on autonomous navigation, dubbed "AutoNav", was formulated. The goal was to bring together, on PC–hardware now available on the market:

1.　The overall software concept for integrating machine perception with the concepts for the planning and execution of missions on all levels, developed by the National Institute of Standards and Technology (NIST, group of J. Albus [33]);
2.　The capabilities of stereo perception in real time, under development by the Princeton-group under P. Burt of the Stanford Research Institute (SRI), running on dedicated hardware [34];
3.　The 4-D approach of UniBwM, with active gaze control and the software packages for dynamic scene understanding and vehicle guidance [20,35].

The final purpose was to demonstrate both with German and US test vehicles the capability of performing a mission on a network of major and minor roads (hardened and natural surfaces) including off-road sections prescribed by sequences of GPS-waypoints in 2001. During these trips, beside obstacles above the driven surface (designated as 'positive' obstacles), 'negative' ones (ditches and large holes below the driven surface) also had to be detected and avoided. In Germany, VaMoRs with a new, 3rd-generation vision system [36–40] was selected for the final demonstration. In the USA, a HMMWV of NIST and of an industrial group, plus a small UGV, were selected as demonstrators. Only the VaMoRs results, with its explorative 'eye' carrying five cameras, will be discussed here. Figure 7 shows the 'vehicle eye' for stereo vision built by UniBwM; it was not intended to be a first version of a practical eye for the more distant future, but a real-world system for testing the software for dynamic visual perception with multi-focal cameras and an intelligent on-board gaze control including very fast saccades.

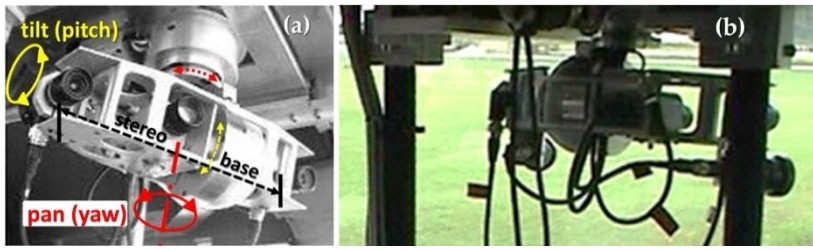

**Figure 7.** Purely experimental, roof-mounted 2-dof vehicle eye for VaMoRs in project AutoNav: (**a**) basic structure with three cameras mounted. (**b**) View from the rear within VaMoRs with the five cameras as used in AutoNav.

In 1997, the third computer system selected for VaMoRs consisted for the first time of four 'Commercial-Off-The-Shelf' (COTS) PC Intel Dual- Pentium-IV, interlinked with a new, also COTS communication network (Scalable Coherent Interface). In 2001 and 2003, VaMoRs was capable of performing the tasks given on the right side in Figure 8 [36–40].

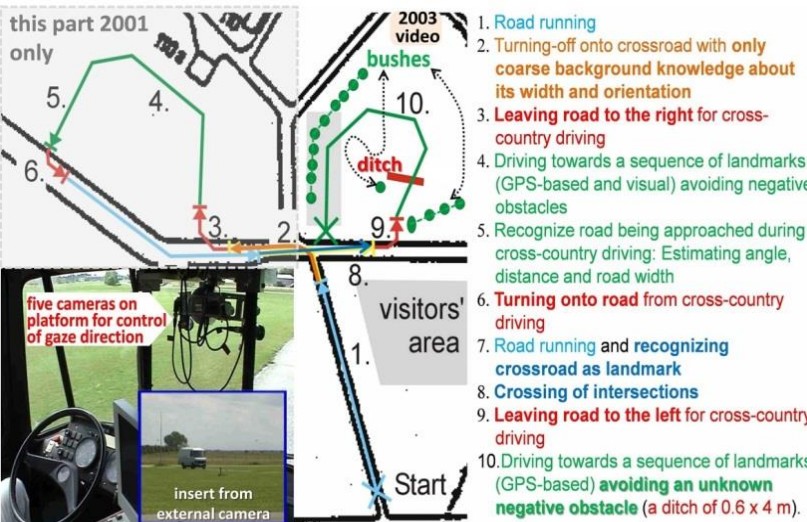

**Figure 8.** Final demo of project AutoNav: Mission performance 'On- and Off-Road' with active (including saccadic) gaze control for turn-offs left and right; detect and avoid negative obstacles.

Point 2 may be viewed in [41] and the rest in [42], a video summarizing the results of project AutoNav. The special 'stereo- computer' of the US-partner for recognizing the exact spatial geometry of a ditch had a volume of about 30 liters in 2001. In 2003, the same mission could be driven using a miniaturized real-time stereo system implemented in the meantime on a plug-in board for one of the PCs. The video film [42] gives an impression of the capabilities of gaze-controllable 'vehicle eyes' for performing the right-hand part of the mission. Again, it was noted that two gaze-controllable eyes, one each at the upper end of frame A of the vehicle, would help in improving flexibility for maneuvers.

Between the computer systems used for these results and future µP for perception in the early 2020s, there will again be a difference in computing power of five to six orders of magnitude [43]. Therefore, besides developing the hardware of a 'vehicle eye', software development for the real-time perception of dynamic scenes will be the major task for the future. As a result of our experience with diverse scenes of autonomous ground vehicle guidance by vision, two cooperating eyes, one at each side above the upper outer corner of the front windshield, seem to be favorable (top of the A-frame).

1.2.2. Another Type of Vision System in This Century

Similar demonstrations as in Germany have been shown in the USA by our partners with their vehicles. The results of all American efforts up to the turn of the century have led to the decision by Congress [44] that "It shall be a goal of the Armed Forces to achieve the fielding of unmanned, remotely controlled technology such that... by 2015, one-third of the operational ground combat vehicles are unmanned." This largely increased the funding possibilities for this new technology of partially autonomous driving. In addition to the activities of the ARL in the general field of research for autonomous driving in unknown environments, DARPA formulated a supply mission in well-known environments, widely exploiting the Global Positioning System (GPS) in connection with precise maps of routes to be driven. These routes were prescribed by sequences of densely positioned GPS-waypoints; the main autonomous part was avoiding obstacles above the surface to be driven. In 2002, DARPA defined such a first supply mission over 142 miles in a semi-arid area in the south of Nevada as the "Grand Challenge" [45]. Obstacle-detection and -avoidance with different types of lasers and with radar was the main issue, since path-following by GPS-waypoints was so tightly prescribed that even hair-needle curves could be handled by a local curve fitting. While the Grand Challenges in 2004 and 2005 focused on the development of autonomous vehicles that operate in an off-road environment with only a limited interaction with other vehicles, the Urban Challenge of 2007 extended this concept

to autonomous vehicles that safely execute missions in a complex urban environment with moving traffic [46].

This goal was promoted since a safe and effective operation in moving traffic is a basic requirement for all missions of autonomous ground vehicles. In November 2007, the CMU-vehicle "Boss" (see Figure 9a) gained the first prize. The arrangement of a large number of diverse sensors all around the vehicle was typical for most contenders (Figure 9b). The second winner, "Junior" of Stanford University, is seen in the center of the figure. All vehicles show that almost all vision-based sensors have been mounted outside the cabin, fixed onto the body of the vehicle.

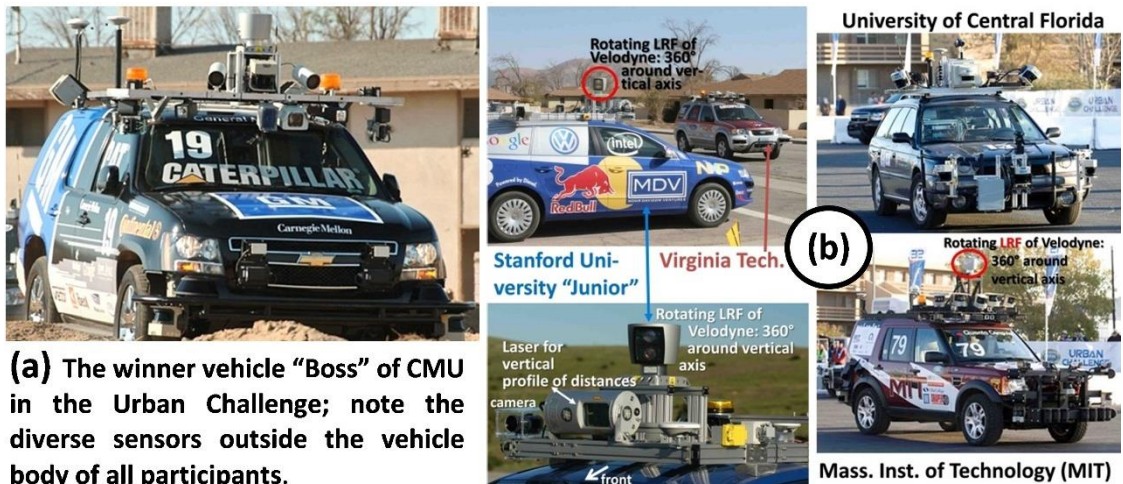

**(a)** The winner vehicle "Boss" of CMU in the Urban Challenge; note the diverse sensors outside the vehicle body of all participants.

**Figure 9.** A collection of images to yield an impression of the amount of external sensors used in the Urban Challenge 2007 for perceiving the environment in short and medium range. Note the revolving Laser-Range-Finder on top of all vehicles.

After the Urban Challenge in the USA, some private companies like Google took the lead in developing capabilities for autonomous driving on roads. Google installed a special institute for the winner of the Grand Challenge 2005, S. Thrun, who also came second in the Urban Challenge with his car "Junior". The institute was provided sufficient funding for equipping up to 10 vehicles for test driving. The rotating Laser-Range-Finder (LRF) with 64 active units arranged above each other, which had been so successful in the Urban Challenge, became the main sensor for obstacle detection. All vehicles had rotating laser-range-finders on top for 360° obstacle detection. In addition, the system relied extensively on GPS data in conjunction with precise local maps. The vision part relied on the stored environmental data of visually predominant stationary objects; these data were not allowed to be older than two days. This set-up differs significantly, in terms of visual perception, from that developed at UniBwM in Germany. While the latter one may be called 'scout-type vision', since it does rely much less on stored data, the former type from Google, as well as most of the others, may be dubbed 'confirmation-type vision' [47]. This, of course, reduces the amount of methods and algorithms needed for object recognition, but it requires a lot of preparatory work for each new application after heavy changes due to weather conditions (like snow) or after catastrophic events (like a hurricane or an earthquake). Most industrial developers of assistant systems for driver support around the globe followed the easier route of a confirmation-type vision.

The exception is the company Mobileye, founded in Israel in 1999, which in 2017 was taken over by Intel Corporation [48,49]. Contrary to UniBwM (with a bi- or tri-focal set of gaze-controlled cameras), Mobileye initially opted for a single camera, mounted and fixed onto the body as a primary sensor for enabling autonomous driving. Due to its capability of interpreting the shapes (like other vehicles and pedestrians) as well as textures and colors of structured objects and areas in perspective projection, it avoided LRF altogether. Early on, the company started developing simple vision systems

on miniature hardware, i.e., proprietary computation cores as fully programmable accelerators. Their system series 'EyeQI', where "I" now runs from 1 (2008) to 5 (2020), covers a wide range of performance levels, also for several cameras and sensors in parallel. The system EyeQ2 allowed input from two cameras (up to 2048 × 2048 pixels) and had the computing devices needed for perception integration on an electronic board (see Figure 10). In 2020, EyeQ5 is predicted to allow input from more than 16 multi-mega-pixel cameras, plus laser and radar; its computational power targets range from 15 to over 200 trillion operations per second, while drawing only moderate electrical power. Another company (NVIDIA) [43] predicts up to 300 tera-operations per second ($3 \times 10^{14}$) in computing power on its boards in the early 2020s. In 2019, Mobileye claimed to have deployed vision safety technology in over 40 million vehicles in connection with many tier-1 automotive companies [48]. They expect ride-share vehicles without a driver (level 5) as a next step soon.

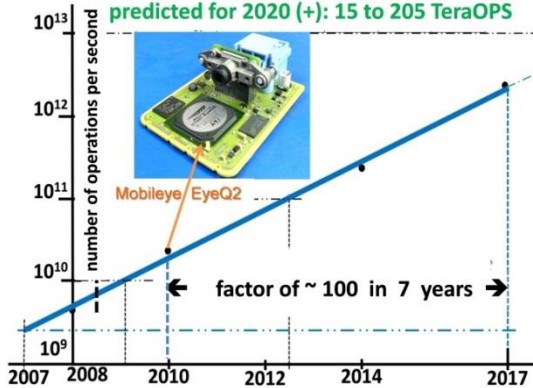

**Figure 10.** Computing power of specific vision systems [48].

With respect to the arrangement of cameras on road vehicles, no new proposals have been noted. For driver assistance systems around the globe, they are mostly mounted at or behind the inside center top of the front wind shield (similar to Figure 11).

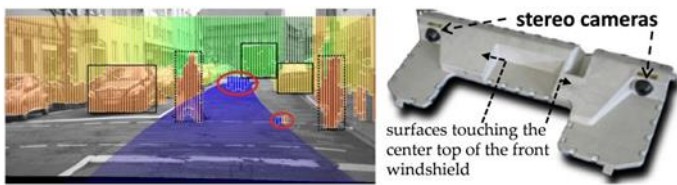

**Figure 11.** Right: Mounting of stereo cameras in the demo-vehicle "Bertha" of Daimler 2013; left: Objects detected by stereo vision with the 'stixel' approach [30,50].

The development in Europe in this century: After the Prometheus project, the European automotive companies concentrated on simple vision systems for road and lane recognition with the new μPs coming along steadily. For obstacle recognition, LRF and radar have been mainly favored. One of the most active industrial groups is that of Daimler. It realized the efficient stereo algorithm 'Semi-Global Matching' on an FPGA in 2008 [50]; Figure 11 shows one result with the demo-vehicle 'Bertha' [30], which after 125 years repeated in 2013—this time fully autonomously—the famous first long-distance drive of Mrs. Benz with an automobile in the year 1888 from Mannheim to Pforzheim.

Almost all automotive companies and tier-1 suppliers in Europe and in Asia started activities in the field of autonomous driving, either on their own or in connection with research institutions. Radar and LRF were widely used for obstacle detection and range estimation, while GPS and precise maps were relied on for navigation; a group of German automotive companies even bought their own company for keeping maps up-to-date and precise. The revolving LRF of Velodyne also found

many customers for precise range estimation all around the vehicle. Figure 12 shows, at the right, an LRF-image derived from 64 laser beams rotating at 10 revolutions per second and covering a range of up to 100 m. It has been taken from the test vehicle MuCAR-4 of UniBwM (a VW-Tiguan, at the center) [51]. The main sensors for the autonomous exploration of previously unknown, unstructured environments are the multi-focal cameras on the gaze control platform active in yaw and pitch (at the left). Together, the two complementing sensors allow for the perception of traversable ground as well as for the detection and recognition of objects relevant for the current mission. This combination is probably the most powerful visual perception system available at the moment.

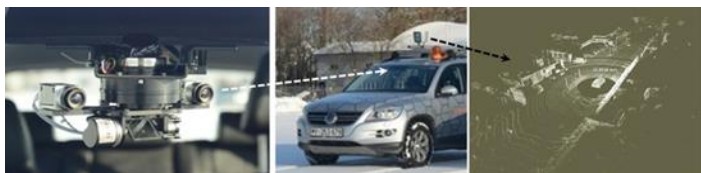

**Figure 12.** Agile multi-focal camera platform (left) as the main sensor, supported by a high-resolution 360° rotating laser scanner mounted on the roof (center). Right: Laser-image in perspective projection. UniBwM/LRT/TAS [51].

The interesting question is whether a vision system (as will be discussed in Section 4) allows one to get rid of the expensive LRF without losing the perception capabilities achieved up to now. It might even be possible to improve the perception capabilities of future vehicles up to the performance level of humans, especially in terms of more distant resolution details for the perception of color and texture.

## 2. Reasons behind the Differences between Biological and Technical Vision Systems

Biological systems have evolved over millions of generations in carbon wetware with the selection of the fittest with respect to the ecological environment encountered; these systems are based on completely different properties of the underlying material as compared to technical vision systems based on silicon (chips). The biological electrochemical units do have switching times in the millisecond (ms) range; the traveling speed of signals in the nerves is in the range of 10 to 100 m/s. Cross-connections between units exist in abundance (1000 to 10,000 per neuron). A single brain consists of up to $10^{11}$ of these units. The main processing step is the summation of the weighted input signals which contain widely (up to now) unknown (multiple?) feedback loops [9].

These systems need long learning times and adapt to new situations only slowly. In contrast, technical substrates for sensors and μPs have switching times in the nanosecond range (a factor of about $10^6$ as compared to biological systems). They can be programmed easily, and they may have various computational modes between which they can switch almost instantaneously; however, the direct cross-connections to other units are limited in number (one to six, usually) but may have very high bandwidths (in the GB/s range). Initially, digital processors were bulky and needed quite a bit of electrical power for running and cooling.

Since the advent of digital μP in the 1970s, the computing power of these units has increased by a factor of ten every four to five years. In [47] (Figure 1), a survey of the transistor count over time from 1980 to 2010 of such μP on a semi-logarithmic scale has been cited; a linear extension of the interpolating line over time suggests that this number may reach the number of neurons in a human brain in the early 2020s. Of course, these numbers are not comparable directly, due to the very different properties of the units; however, they may indicate that, with respect to complexity, similar overall systems should soon be in reach on a technical basis. Up to now, the computing power of μP-systems has increased by more than a factor of 1 million since the early 1980s, while at the same time the volume and need of electrical power has decreased dramatically.

Thus, computing power does not seem to be a limiting factor any more for the future. Therefore, we will concentrate on system design and computer software, taking the facts mentioned in the introduction into account.

## 3. Conclusions for the Design of a Technical Eye of High Performance for Automotive Applications

The partially separated processing of image data both in the eye and in the visual cortex of the human vision system based on carbon molecules is not mandatory for technical vision systems based on silicon molecules. Sensors and computers may be kept separate. In fact, several high-resolution images may be communicated in parallel from the sensors to the computing elements; however, the multi-focal, saccadic type of image sequence analysis may be of advantage in technical systems too. Requirements that need to be fulfilled are outlined next.

In the near environment of an autonomous vehicle, a large FoV should be covered. If the road being driven and a road crossing have to be viewed simultaneously, 110° to 120° seem to be adequate. With the number of sensing elements limited by their size and the space available in an eye (or on a camera chip), the resolution of the imaging sensor is bounded. The highest resolution required for object-detection, -recognition and -tracking at larger distances may lead to a large number of sensor elements. Therefore, the locations and the potential viewing directions of all cameras on the vehicle are essential design parameters. Are many cameras with a small neighboring FoV, possibly collected group-wise together as 'eyes' and mounted in a fixed way onto the body of the vehicle, a good solution? In the frontal range from −115° to +115° relative to the vehicle heading, the highest resolution corresponding to 0.2 mrad per pixel in the image should be available. {General remark: With respect to the localization of intensity edges, the high-resolution results should correspond to the capability of the human eye (0.2 mrad in all directions at all times).} This leads to 20,000 pixels covering the frontal angular range of 230°. However, this high resolution is not required all the time in all directions. Usually, in road traffic, the highest resolution is needed down the road ahead in the driving direction. Approaching cross-roads, it is desirable to have this high resolution available for small periods in time over the complete FoV, while less resolved images may suffice between these periods. This will be discussed in Section 4.1 for two frontal 'vehicle eyes' mounted in a fixed way at the upper end of the left and right of the A-frame of the vehicle.

Since highly resolved images are only needed for very small regions of the total FoV, the question arises of whether a better approach is to collect these huge amounts of data only when and where they are actually needed. This has led to multi-focal saccadic vision in biological systems. Of course, this complicates the software development for technical dynamic scene understanding. However, including spatiotemporal models for the interpretation of image sequences right from the beginning in feature extraction, as done in the 4-D approach since the early 1980s [16], has shown to be very efficient. The volume of data to be handled may be reduced by orders of magnitude.

Experience with Hardware-In-the-Loop (HIL) simulations and with the UniBwM-test-vehicles VaMoRs and VaMP in the 1990s has shown that a joint evaluation of the multi-focal image sequences with saccadic changes of the gaze direction is well suited for technical dynamic vision too. As one of many alternative designs for active gaze control, a tri-focal vision system with focal lengths separated by factors of 3 and 4 will be discussed in Section 4.2.

## 4. Alternative Lines of Development

The answer to the question "which is the preferable approach to follow" depends on the field of application. Are the vehicles exposed to fast and heavy angular perturbations leading to motion blur in the images under unfavorable lighting conditions? If so, gaze control for image stabilization is necessary anyway; the question is then: what is the best angular range of the gaze control (amplitudes and rotational rates). Today, the prevalent opinion in the automotive industry is that motion blur is no issue in modern imaging sensors for today's applications. Therefore, version-1 of a technical eye is here

considered, with all cameras mounted in a fixed way onto the body of the vehicle. The cameras that are actually available may then be very small and light-sensitive; they may also have a high dynamic range with respect to the light intensity. Whether very small cameras of the future will have sufficiently high capabilities in these areas at only slightly higher costs than standard cameras will be a deciding factor between the two types of technical vision systems considered next.

## 4.1. Eyes with Cameras Mounted in a Fixed Way onto the Body of the Vehicle

The simplest extreme solution would be mounting many cameras in a fixed way onto the vehicle body wherever required (see Figure 5). If average human visual acuity is the goal, a camera with 4000 pixels per line will cover about 46°; five cameras will be required to cover a FoV of 230° horizontally (see Figure 13). With a requested vertical FoV of about 45°, a set of five cameras is needed. Two eyes are chosen here, one each at the upper end of the A-frame at each side of the front windshield (see Figure 14). In order to allow coverage of the sides of the vehicle, the axis of symmetry of each eye is rotated by 46° away from the center line of the vehicle. Thus, the left and the right eye are identical with respect to their structure, but are mounted differently. Since only the near environment is of interest to the oblique sides (away from 0° (ahead) and 90° across the vehicle), the two outer cameras of the arrangement may have a reduced resolution, shown with light-gray longer-dashed lines in Figure 13.

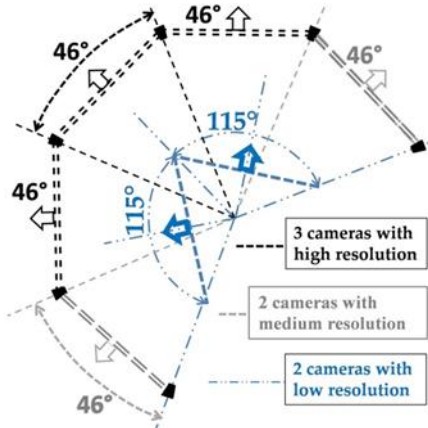

**Figure 13.** A potential technical eye mounted in a fixed way on the vehicle body; it has seven cameras with three resolutions.

The arrangement chosen allows a single circular arc of sensors around a vertical axis, mounted in a fixed way onto the body of the vehicle. If cameras with a 2000 × 2000 pixel resolution should be much less expensive or smaller, the same FoV as in Figure 13 may be covered by 20 cameras (instead of the five here). Vertical pairs would be arranged around the vertical axis, each pair rotated by 23° relative to its neighbors. This would yield less distortion in the areas of transition between adjacent images.

For an efficient understanding of automotive scenes it is important to start the computational analysis with low-resolution images from the regions nearby (at the bottom of the image). Therefore, when using only high-resolution sensors (without the ones painted in blue in Figure 13) it makes sense to compute several (2 × 2)-pyramid levels to start with. The next higher pyramid level is computed by forming one pixel on this level by averaging four pixels in a square on the lower level. This reduces the amount of image data on the second pyramid level by a factor of 16. Starting from 0.2 mrad/pixel, on level 2 the resolution will be 0.8, and on level 4 it will then be 3.2 mrad/pixel. Level 4 means that at a 10 m viewing range, a lane marking of 12 cm width will be covered by three to four pixels, a reasonable number for detecting the orientation of an edge precisely. A car with a body-width of 1.8 m will be covered by about 60 pixels, sufficient for recognizing some details, e.g., the black tires underneath being four to eight pixels wide (12 to 25 cm). The number of pixels on this pyramid level, however, is less than 0.4% (a factor of 1/256) of the original high-resolution level.

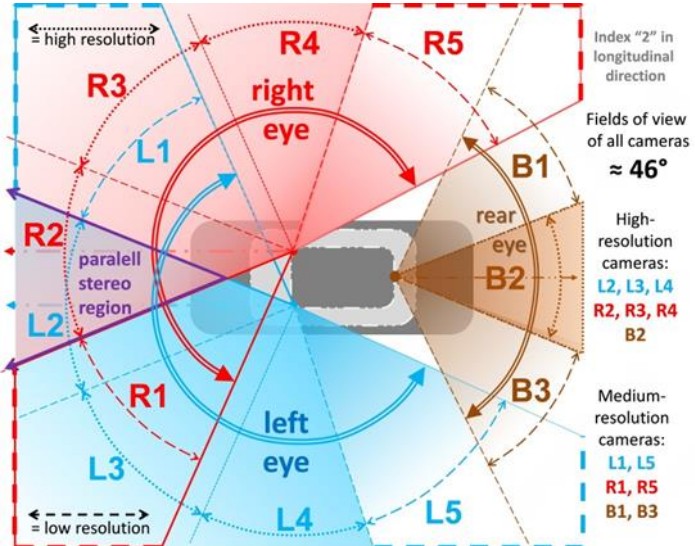

**Figure 14.** Car with a pair of body-fix eyes in the front at the left (Li) and right side (Ri) of the roof, and a set of three cameras looking to the rear (Bi).

Since the computation of pyramid images requires quite a bit of computing power, it seems worth considering whether directly sensed low-resolution images, as sketched in blue color in Figure 13, could be of advantage; in any case they would provide redundant images, increasing safety. With 1_K pixel per row, an angular range of 115° can be covered with a resolution of about 2 mrad/pixel. Thus, two cameras with a side ratio of 16:9 will cover a FoV of about 2 × (115° × 65°). The number of pixels per frame for the two cameras is then 1.125 megapixel (MP). At 40 m distance, one pixel of this camera covers about 8 cm normal to the line of sight. The rear view of the lower body of a car of size 1.8 × 0.9 m at that distance will be covered by about 22 × 11 pixels. This is reasonable for alerting interest in special sets of features in road scenes. Compared to the total number of pixels resulting from a design as shown in Figure 13 on the high-, medium- (the second), and low (the fourth) pyramid levels of higher-resolution original images (53.6 MP), this means a reduction in the data volume by a factor of almost 50.

Under standard conditions, only this reduced data volume is analyzed. The medium resolution of 0.8 mrad/pixel (on the 2nd pyramid level) means that at a 50 m distance one pixel covers an area of 4 × 4 cm normal to the line of sight. This is sufficient for lane detection up to about a 70 m range. At the fourth pyramid level, one pixel covers 16 by 16 cm at a 50 m distance. This may be sufficient for detecting larger objects with contrasting surfaces; even four to six pixels often suffice for detecting dominant features of objects, e.g., a large difference in brightness. In case more information on such a set of features and its environment is required, the higher-resolution image data of this region (say two low-resolution pixels added to each side around the center of the point of interest) need only be transferred for a more extensive analysis.

Thus, an object of size 6 × 3 pixels on the fourth pyramid level would trigger the transfer and analysis of a sub-image of size 1120 pixels of the medium-resolution image around the detected center. Here, the feature analysis will indicate whether an even finer resolution is needed and in which subarea. For the four medium-resolution images of both eyes with indices 1 and 5 in Figure 14 (dashed circular double-arrows), the data reduction per object (set of features) is two to three orders of magnitude when using two pyramid stages. For the entire front hemisphere from −115° to +115°, high-resolution image data are stored and selectively available on demand.

The total FoV covered horizontally by both eyes ranges from −161° to +161°. In the frontal sub-region with index 2 (marked in magenta in Figure 14), a binocular stereo interpretation is possible. Using the 4-D approach for dynamic vision [16], these regions could be adapted in size and predicted

over time for the next image in order to directly track the object in the high-resolution image sequence. Computing any pyramid data on higher levels does not need to be done in this case.

In order to sum this up: Figure 14 shows three cameras "Bi" looking to the rear (in brown color), in addition to the multiple fields of view Li and Ri of the five cameras at the front of a car. Two eyes, one each at the left- (L in blue) and right-hand side (R in red) are shown. In the figure, a high-resolution camera B2 for the back side has been chosen for the center (from −23° to +23°). Medium-resolution image data are available to the rear sides (indices 1 and 3). Thus, the rear hemisphere is covered from −69° to +69° if the same cameras are used as in the front (for simplicity of service); of course, B1and B3 could be selected with a larger FoV in order to cover the entire rear hemisphere. However, the gain would only be the two small white triangles to the side.

What amount of data and computational operations is needed to visually track ten objects in parallel, following either the pyramid approach or separately sensed image data on three focal levels? A rough estimate shows a reduced load by a factor of two to three orders of magnitude is required to provide the proper high-resolution image data to the processors for the scene evaluation. This indicates that an early transition to multiple hypothesized single objects moving in the observed scene considerably enhances the efficiency of dynamic scene understanding. If the two additional low-resolution cameras are chosen to avoid the need for the computing image pyramid levels, and if one of the focal spacings is reduced to 3 (yielding a total of 12 instead of 16), this lessens the number of operations to about 0.2% of those for all higher-resolution images. The positions of the corresponding regions for ten objects may, of course, change from frame to frame within an FoV of 230° × 46° (see Figure 14, regions with indices 2 to 4). By accepting small delay times for saccadic gaze control, the reduction in data volume can be further improved, even with higher capabilities in resolution.

### 4.2. Gaze-Controllable Eyes with Tri-Focal Saccadic Vision

Assuming that developments in the fields of tiny video cameras and corresponding gaze control platforms will continue to deliver smaller units, the camera specifications chosen for the design considered are more likely to be conservative for the long-term developments in automotive sensing. About 5 cm (or less) seems reasonable as the diameter of a 'vehicle eye'. The capability of very fast changes in gaze direction to sets of interesting features allows for the reduction of the data rates, as discussed above. This may be achieved with gaze control at the expense of small delay times until scene understanding is achieved on the interpretation level. For humans, this delay time is in the order of 0.3 to 1 s, i.e., >7 video frames at 25 Hz (>9 at $33^1/_3$ Hz). This is well achievable with technical perception systems [16]. If the maximal resolution is requested to be the same as that assumed above (0.2 mrad/pixel), and if the lateral range to be covered at a 200 m distance is 32 m (about the width of a four-lane highway in both directions), the number of pixels needed per line is 800; they cover an angle of about 9°. A lane marking of 12 cm width is covered by three pixels, and the lower body of a car of size 1.8 × 0.9 m is covered by about 1000 pixels in total. This resolution also allows for good recognition of the lower parts of the wheels (16 to 28 cm wide) seen from the back [52,53]. These features at a specific distance to each other below a uniform area allow for the recognition of a car as opposed to a big box.

With the active control of the gaze direction in the range of 150° horizontally (90° away from the car, 60° over the car body) and 30° vertically, a tri-focal set of four cameras (see Figure 15) allows the same high-resolution FoV as discussed in Section 4.1, even in a larger total region. While for automotive applications the low and medium resolution cameras may have rectangular image sizes, for the high-resolution camera a quadratic image with about 800 pixels in each direction may be sufficient. The corresponding data volumes and rates are given in Table 1. Here, the data rate for one object tracked at all resolutions is 0.21 GB/s (for one eye with full redundancy on three levels of resolution). Using the same set of two low-resolution cameras for the gaze-controllable eye, their data rates are 0.054 GB/s (at 25 Hz; 0.072 GB/s for $33^1/_3$ Hz). These cameras are mounted such that at a gaze direction of 90°, their FoV covers the entire side of the vehicle; this yields their gaze directions of ±32.5° to

both sides of the main gaze direction of the eye. As a consequence, the front region is covered in a multiple-redundant way by two low-resolution cameras and one medium-resolution camera of each eye in a FoV of 50° (marked orange in Figure 15).

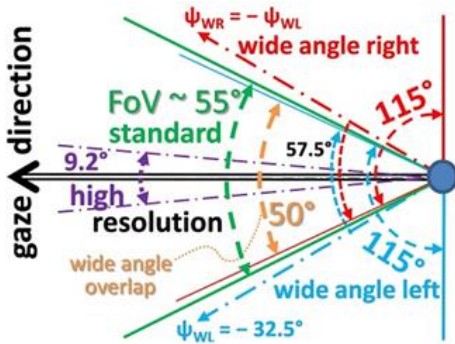

**Figure 15.** 'Vehicle eye' with a tri-focal set of four cameras: two times 115° FoV with low resolution, 55° with medium and 9.2° with high resolution.

**Table 1.** Parameters of a gaze-controllable 'vehicle eye'.

| Type (Resolution) | Low | Medium | High | Remark | |
|---|---|---|---|---|---|
| Fields of view (in °) | 115 × 62 | 55 × 31 | 9.2 × 9.2 | Left (−) and right (+) for 'low' | |
| Imaging characteristics (resolution) | 2.4 $^1/_4$ of med. | 0.6 $^1/_3$ of high | 0.2 | mrad/pixel, acuity of edge localization | |
| Pixels/line | 800 | 1600 | 800 | These are rough estimates according to a pinhole model | |
| Number of lines | 450 | 900 | 800 | | |
| Data volume/frame | 2.16 MB | 4.32 MB | 1.92 MB | 3 Bytes/pixel; sum = 8.4 MB/cycle | |
| Data rate at Hz: | 25  0.054 GB/s $33^1/_3$  0.072 GB/s | 0.108 0.144 | 0.048 0.064 | sum in GB/s | 0.210 0.280 |

Choosing a high-sensitivity black-and-white camera for low resolution (large field of view) and color cameras for medium and high resolution would be closer to the capabilities of the human eye. Delay times for saccades of up to 40° are in the range of a few standard video cycle times (around two to three tenths of a second). In summary, by choosing properly designed gaze-controllable 'vehicle eyes', the amount of image data to be handled may be reduced drastically. Figure 16 shows typical images scaled at a factor of three (upper two images) and four (lower two images) apart; they were taken towards the end of the last century with VaMP and with the camera set inserted in the upper image.

Here, color cameras had been selected for the low and medium resolution, while a highly sensitive b/w-camera was used for the high resolution. The details resolved in the upper image are impressive when comparing this region with the rectangular sub-images marked in the lower two images. Medium resolution is good for object recognition that is not too far away, and low resolution fits best for recognizing the entire situation. High resolution is needed in a few smaller regions only. Notice that the license number of the second vehicle in front and even a phone number written on its body are well readable.

For an object of size $n_{ob} \times m_{ob}$ in a low-resolution image, extending the search region by two pixels to each side, the resulting region in the high-resolution image is $(n_{ob} + 4) \times (m_{ob} + 4) \times 12 \times 12$ pixels around the center. An example with size $6 \times 3$ low-resolution pixels for an arbitrary object yields a data volume of 10,080 pixels/frame and a data rate from a high-resolution search region of 0.756 MB/s.

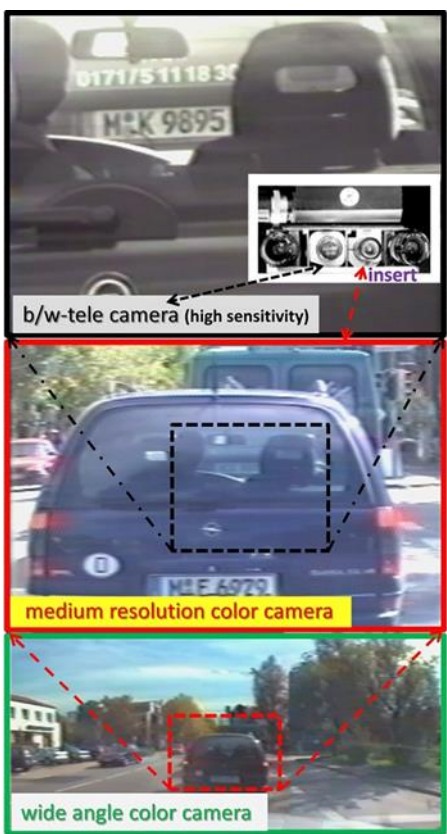

**Figure 16.** Tri-focal images of a traffic scene; VaMP 1999 (cameras as insert).

Instead of losing performance in perception compared to cameras mounted in a fixed way, gaze control allows for a direct compensation of rotatory perturbations by feedback from signals from inexpensive inertial sensors on the pointing platform. Figure 17 shows a reduction in amplitude by more than an order of magnitude in pitch for a braking maneuver of VaMoRs. In addition to this, via angular rate commands to the gaze control, the tracking of fast-moving objects can be done for a reduction of motion blur; this is especially valuable under low light conditions. In general, it increases the competence in perceiving dynamic scenes. In special application areas, this advantage may be decisive; whether it is favorable for road vehicles as well has yet to be seen.

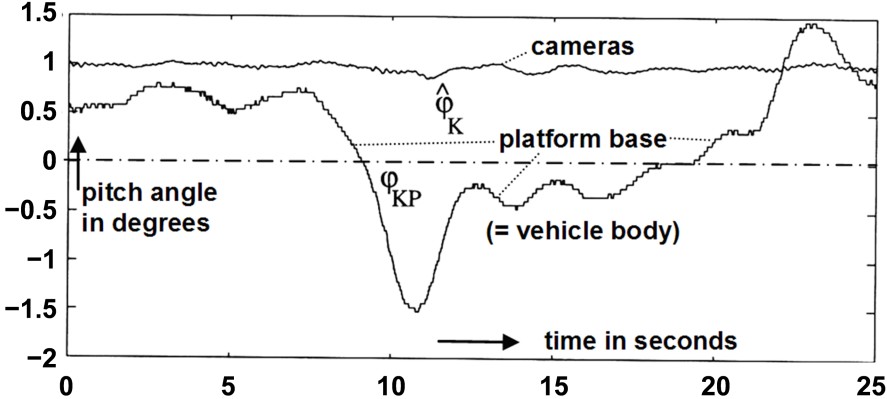

**Figure 17.** Inertial gaze stabilization for a braking maneuver with VaMoRs.

Even small angular perturbations on the vehicle of 1° per frame (40 ms at 25 Hz) shift the corresponding input-ray of a body-fix sensor to a location about 3.5 m perpendicular off to the side at 200 m distance; this corresponds to the same point in the real world being shifted by about 87 pixels

laterally in the high-resolution image. Due to the small aspect angles on the longitudinal axis, the effect on the depth perception is more pronounced. This would cause large search activities in the next image in order to understand the image sequence. With the 4-D approach, good predictions for gaze control can be made by taking both object motion and ego-motion systematically into account.

### 4.2.1. Considerations with Respect to Levels of Resolution for 'Vehicle Eyes'

Though animals with more than two eyes can been found, two eyes with the capability of fast gaze control predominate by far in the biological realm. This allows stereo vision in the near range and redundancy in case one eye fails. In combination with active gaze control, the contradictory requirements of resolution on the one side, and the number of cameras needed on the other side, are resolved by choosing more complex imaging sensors (see Figure 1 for the biological realm). Vertebrate eyes have areas of lower resolution over wider fields of view and (very) high resolution in the central area. Usually, the transition is gradual. In the technical example discussed here, three discrete levels of resolution are chosen in order to use three types of cameras with a constant resolution in their FoV. The corresponding object discovered with a low resolution may then be analyzed with an image resolution four or even 12 times higher than needed just for detection. The discrete values of the camera parameters chosen here are not meant to be the optimal ones; these depend on the criteria used. The values chosen are just reasonable numbers based on our experience with multi-focal technical eyes. It has been discovered that the recognition of objects after a saccade is more efficient if the focal spacing is not larger than by a factor of three to four. Via feedback of prediction errors for optical features to the control of the gaze direction, the interesting set of features may be automatically tracked, thereby reducing motion blur and allowing for better object recognition and tracking.

Let us assume that a highly sensitive video sensor to be used will have 800 pixels per row. If the requested total FoV of the low-resolution part of the eye is 115°, one pixel in this array will have a lateral coverage of $\approx 2.4$ mrad; the corresponding lateral ranges in cm at six typical distances for automotive applications are given in row 3 of Table 2.

**Table 2.** One pixel covering $\delta$ mrad of angle means a lateral range in cm at a distance $L_{x\,m}$.

| Distance $L_{x\,m}$ in m | | 2 | 6.25 | 12.5 | 25 | 50 | 100 | 200 |
|---|---|---|---|---|---|---|---|---|
| angle $\delta$ in mrad | | Lateral coverage in cm of 1 pixel normal to line of sight | | | | | | |
| low | 2.4 | 0.48 | 1.5 | 3 | 6 | 12 | 24 | 48 |
| medium | 0.6 | 0.12 | 0.375 | 0.75 | 1.5 | 3 | 6 | 12 |
| high | 0.2 | 0.04 | 0.125 | 0.25 | 0.5 | 1.0 | 2.0 | 4 |

Perpendicular to the line of sight, one low-resolution pixel will cover about 48 cm at a 200 m distance; road vehicles seen from the back will thus be covered by about 6 to 12 pixels. If their brightness is quite different from the surroundings, this is sufficient for detecting that there may be something of interest; however, no recognition is possible with this low resolution. If the highest resolution is requested to be 4 cm (0.04 m) per pixel perpendicular to the line of sight at the distance $L_{0.04} = 200$ m, the focal partitioning given in the second column of the table results. The medium-resolution camera with 1600 pixel per row (column 3 in Table 1, and row 4 in Table 2) will be able to see a traffic light located 2.5 m to the side at distances above about 5 m when looking along the centerline of the first lane of a road. It allows for the detection of a traffic light nearer than 100 m in range. The low-resolution camera covers a single traffic light of 0.2 m diameter by about 9 pixels below a about 25 m range. If the high-resolution camera tracks the traffic light, the two low-resolution cameras remain capable of mapping the entire traffic situation in front of the vehicle nearby. The gaze direction during this fixation will change similar to the upper curve shown in Figure 6. However, traffic lights on the different types of roads in cities and around rural roads may be found in a much larger environment of the road (on both sides and even high above the road). When moving towards the lights, this has shown to be a

challenge for cameras mounted in a fixed way onto the body of the vehicle. This capability of fixation onto an object, either moving itself or due to ego-motion (or both together), is one of the big advantages of active gaze control.

### 4.2.2. Number of Gaze-Controlled Eyes Needed

With a horizontal field of view that is two times 115° for the low-resolution cameras of a single eye (see Figure 15), and a range of potential gaze directions in azimuth from −90° (left side) to +60° (right side of Figure 18) for the left eye (−60° to +90° for the right eye), most of the interesting environmental regions around a vehicle can be covered with two eyes when properly located. In the central range of ±60° around the heading direction of the vehicle, even redundant coverage with a high resolution is possible by controlling each eye separately (called vergence). The part (H) at the center is the standard orientation straight ahead, as given in Figure 18; the part ('Side', at the left) shows the extreme yaw angle away from the body for the left eye. For this gaze direction, the low-resolution (wide-angle) camera to the outward side of the eye covers the entire side of the vehicle (shown in blue color, also in Figure 19).

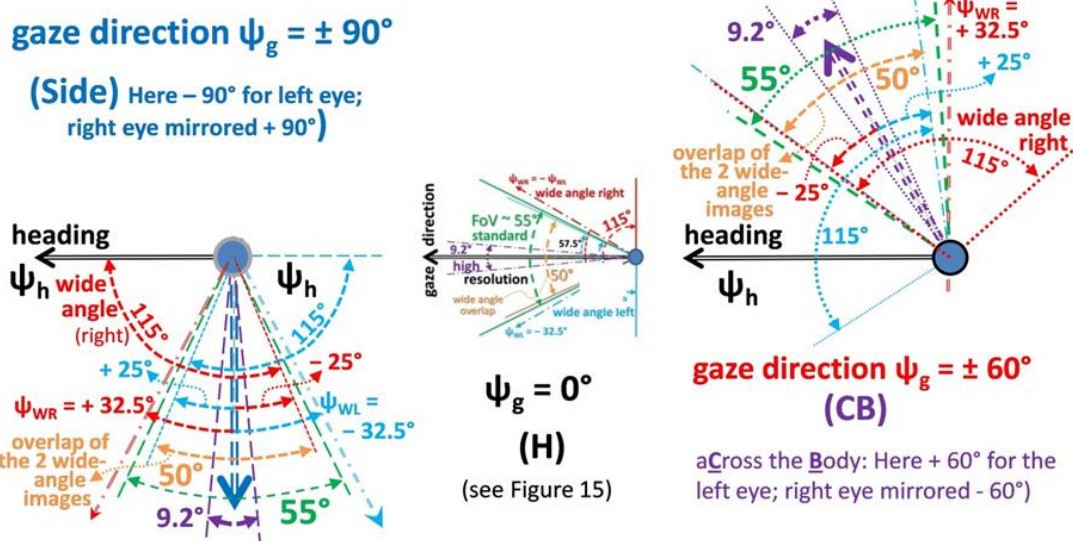

**Figure 18.** Trifocal gaze-controlled (left-side) 'vehicle eye' with four cameras: (Side, at left): Looking −90° to the left (relative to the heading direction of the vehicle); (H) miniaturized Figure 15 looking straight ahead; (CB, at right) extreme gaze direction in azimuth across the body is + 60°. The eye on the right-hand side is vertically mirrored (+90° and −60°).

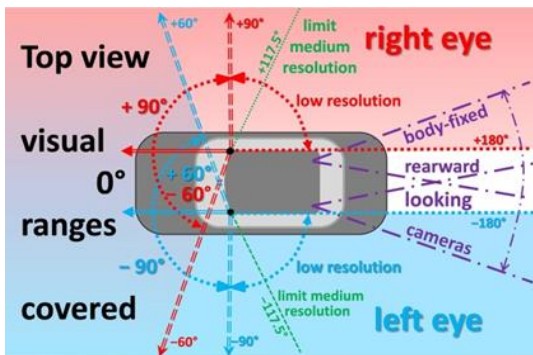

**Figure 19.** Coverage of the environment with two gaze-controlled tri-focal eyes for the front hemisphere and two body-fix cameras for the rear one.

As the wide-angle cameras are mounted so that the outer side of the image ends just at 180°, both cameras have a region of central overlap of 50° in the gaze direction of the eye (orange in Figure 18). The two eyes should be positioned at the upper end and outside corner of the front windshield of a car; the potential FoV are shown in Figure 19 together with two FoV of cameras mounted body-fixed near the upper outside end of the rear windshield. Due to the curved shape of the roof of the car, the gaze directions to the front across the body are generally limited to about 60°. The large FoV of the wide-angle cameras (115°), mounted with their line of sight under an angle of $\Psi_W = \pm 32.5°$ to the side of the gaze direction $\Psi_g$ of the eye, allows for coverage of the entire side of the vehicle with the corresponding eye at a gaze direction of 90° (see Figure 19). These large angles are only of interest nearby for the low-resolution cameras in order to perceive objects in the neighboring lane. Objects further away on a crossroad may be imaged by the medium-resolution (standard) camera up to a maximum of 117.5° (green lines in Figure 19), and with the high-resolution camera up to 94.6°. The medium-resolution camera is sufficient for perceiving objects that are not too far away on a crossroad in greater detail.

By installing one of these eyes at the upper end at each side of the front windshield, the front hemisphere will be steadily covered with low-resolution imaging when looking straight ahead. With gaze control, a region of 235° may be perceived with medium resolution with at least one eye (separately controllable). That means that each interesting set of features discovered in a FoV of ±90° at low resolution in a coordinated "surveillance mode" with the gaze direction straight ahead may then be analyzed by the medium-resolution camera in a range of 235° within a small fraction of a second due to the saccadic gaze control.

For smaller gaze angles (say $\Psi_g = \pm 45°$), both eyes may be directed towards the same real-world object, thereby allowing 4-D stereo perception by exploiting the extended vergence control over some time. With a stereo base of about 1.2 to 1.7 m (width of the front windshield), these depth perceptions should be reasonably good up to distances of at least 20 m. However, in the very near range the large stereo base may cause some difficulties due to the different aspect conditions for parts of the object.

With an individually controllable tri-focal eye on each upper side of the front windshield, only the rear hemisphere in the central part is not perceivable (white area in Figure 19). For the very limited types of maneuvers performed when driving backwards, body-fixed cameras, widely in use today for backing up, may also be sufficient in the long run, especially when additional radar sensors are used for detecting distant vehicles closing in. Two medium-resolution cameras mounted in a fixed way at each upper side of the rear windshield would be preferable for more precise observations.

To summarize, it is emphasized again that in the region of ±117.5° the gaze-controllable eyes allow the tracking of objects of special interest for the precise perception of a situation by reducing motion blur. In addition, these objects may be analyzed simultaneously at different levels of resolution, which may allow for a more stable interpretation. Additionally, keep in mind that monocular stereo perception is possible with the 4-D approach when moving [16].

### 4.3. Car Design for Both Types of Eyes

The two types of 'vehicle eyes' discussed above will most likely not be developed in the near future, since they are not needed for the low capabilities required for driving in well-known and smooth environments (mainly: known roads) with support from GPS as well as from detailed high-precision maps that are kept actual. However, after some catastrophic event (e.g., earthquakes, weather extremes, etc.) or in unknown environments, vehicles with more capabilities would be advantageous. This is also valid for vehicles exploring other planets or moons in astronautic missions. Scout vehicles in the defense realm or for fire brigades are other examples. Once these high-performance vision systems are available and can be realized at relatively low costs, their migration into the market of general vehicles cannot be excluded. Therefore, they should be conceived as a separately marketable sub-system right from the beginning.

Since one can hardly predict which of the approaches discussed here will be superior from a cost–performance point of view in a few decades, it seems reasonable to design the changes necessary at the upper end of the A-frame of a car so that it fits for both types. Figure 20 shows one possible design of the upper end of the A-frame of a car, providing a potential standard solution for mounting front eyes.

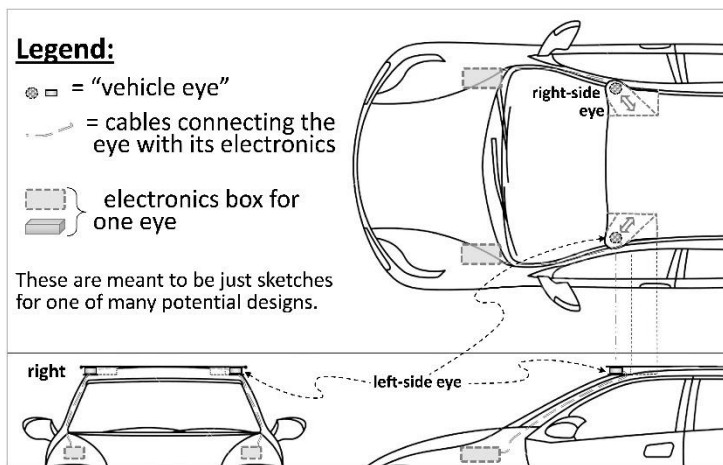

**Figure 20.** One potential design of the upper end of the A-frame for a standard solution for mounting front eyes.

A section of a cylindrical glass tube for protecting the cameras, together with a wiper for clearing the view (see top of Figure 21), and a lid for covering the glass cylinder are part of the adapted design of the roof. The position for the wiper at rest is in the back corner of the glass cylinder. Of course, water supply for cleaning the glass cylinder has to be included. The eye itself, with all connections needed for mechanical and electronic functioning, should be mountable from the inside of the car, if possible as a compact unit (indicated by diagonal arrows, top right in Figure 20). The hardware of the electronics (possibly with computing power for feature extraction from the wide-angle images) may be located near the lower end of the A-frame to keep the cable lengths short. Eventually, when interesting features are detected, a quick change in the gaze direction may be initiated directly from here with little delay time and with corresponding information sent to the central unit for perception. The cables to the eye may run well protected in the inner side of the A-frame.

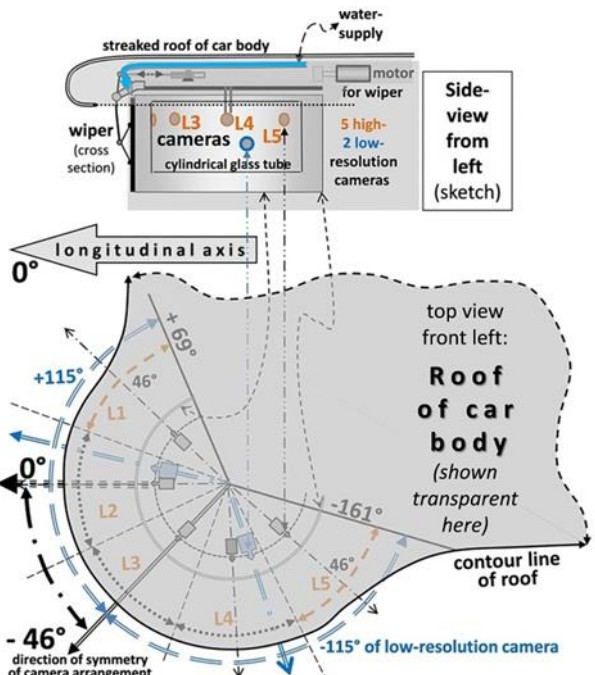

**Figure 21.** Body-fix left-side eye: The specific FoV as given in Figure 13 are shown in the lower part (by making the roof transparent here). The two low-resolution cameras (in blue) cover the same FoV as all higher-resolution cameras together (230° from −161° to + 69°); their images meet at the gaze direction of −46°. The top part sketches a view from the left-hand side; the cameras are inside a cylindrical (semi-transparent) glass tube that is kept clean by a wiper, whose rest position is at the right corner.

### 4.4. Sketch of an Eye Mounted in a Fixed Way onto the Body of a Car

One possibility for the arrangement of cameras in such an eye has been discussed in Section 4.1 with many cameras collected in one unit and mounted in a fixed way onto the vehicle (Figure 13; Figure 14). For best viewing conditions, it should be mounted as far as possible above the ground. Figure 21 shows a favorable configuration for the left eye of a car. With the vanishing costs of cameras, it seems reasonable to add the two cameras with low resolution but with the same FoV as those needed for high resolution (see the blue parts of Figure 19 and the bottom of Figure 21). This provides redundancy in environmental mapping and thus safety. In the case of a failure of one or more of the high-resolution cameras, these low-resolution images will suffice for slow autonomous driving to the next service station, if the corresponding software is available.

Figure 21 shows a sketch of such a vehicle eye without gaze control positioned at the upper end of the left A-frame of a car: The lower part outlines the arrangement of three high-resolution cameras (L2 to L4), two medium-resolution cameras (L1 and L5) and two low-resolution cameras in blue color. In the FoV of the cameras L1 and L5, only nearby objects are of interest, so that the overall data volume can be kept low (a reduction from 32 to 2 mega-pixels, or a saving of 90 MB of data per cycle; this reduces the data flow rate of the five cameras by about 37%).

The eye at the right-hand side of the car is mirrored by a vertical plane in the longitudinal axis through the center of the vehicle, enabling binocular vision in the frontal range from −69° to +69° only. Both eyes together allow a binocular stereo image interpretation with a relatively large stereo-base ($L_{StB}$) of $L_{StB}$ = 1.2 to 1.7 m (depending on the width of the vehicle). Crossroads and the nearby lateral vicinity (up to ±161°) can be seen by one eye (left or right) only. The low-resolution cameras, however, provide redundancy for the perception of lower details. Assuming a sufficient accuracy in the distance estimation of up to about 15 × $L_{StB}$, a separate sensor (e.g., LRF) for good viewing conditions does not seem necessary; a radar is needed for all weather conditions anyway.

A glass structure for protecting the cameras is mandatory; a section of a cylindrical tube is chosen here. One design possibility is sketched with the position of the wiper in action in the front (top left); at rest, the wiper would be parked in the rear outer corner. When the eye is not needed, a lid may cover the glass tube. These supplementary devices will have to be part of the adapted roof design (see Figure 21 top).

### 4.5. Sketch of a Gaze-Controllable Eye with Tri-Focal Saccadic Vision

In contrast to a body-fixed eye, a tri-focal, gaze-controllable eye provides redundancy on three levels of resolution for the same region by separate cameras with considerably reduced data rates (−94.4% ≈ 1/18). Image stabilization on all three levels is achieved by inertial feedback of external perturbations onto the gaze control platform. Figure 22 shows a simple-minded design as an extension of the line of development of the larger platforms tested with the vehicles of UniBwM. They had separate degrees of freedom in pitch, marked here by (1) at the top, and in yaw, marked in red by (2) for the motor in the bottom of the unit. The presentation is given here for the maximum gaze direction to the outside (−90° to the left). In this design, the cables for power supply and for transferring the sensor signals have to allow angles of rotation in yaw of 150° and of about 50° in pitch. These cables and the bearings for the axes of rotation may be causes for wear and tear, needing service every now and then.

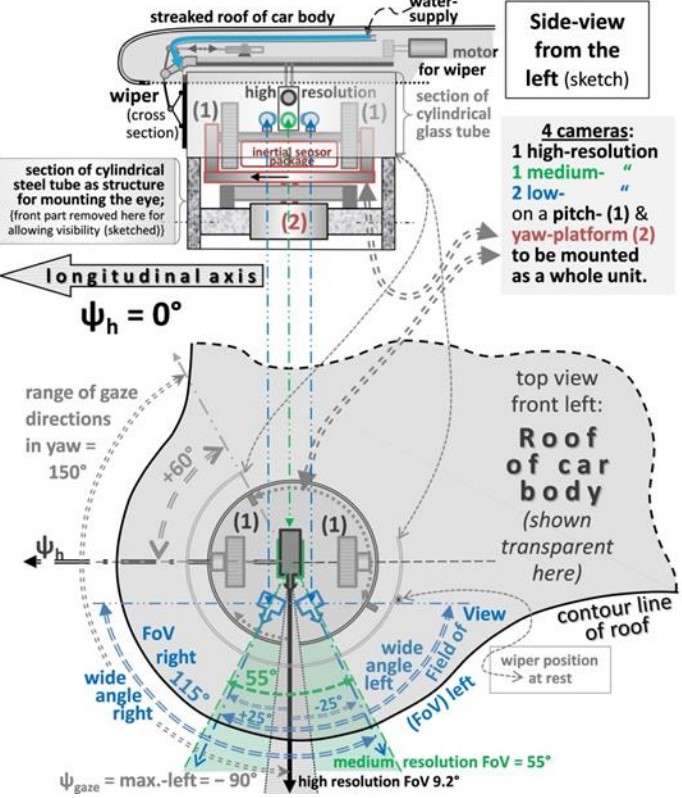

**Figure 22.** Gaze-controlled 'Vehicle Eye' (left side) at the upper end of the A-frame capable of providing tri-focal images in the same FoV as the body-fix eye in Figure 21; however, it has a data rate lower by two orders of magnitude, but with a delay time of a few video-cycles.

An ideal design for such an eye would include a (magnetic) levitation and torque application in the two axes of motion by electromagnetic fields, avoiding mechanical contacts. For sensor data transmission from the eye to the world outside and for power supply to the sensors, a solution without material links would also be ideal. Whether this will also remain impossible in the long run (especially the combination of both) is hard to predict. Levitation and control of motion alone, without a mechanical touch, could make a gaze-controllable eye very attractive. According to the data given

above, data rates are one to two orders of magnitude lower than for the eye mounted in a fixed way onto the vehicle. Thus, a gaze-controlled tri-focal eye may deliver all the visual information required, with the needed computing power much reduced and, in addition, with the information redundantly obtained from different cameras. Future eye-designs may reduce the delay time for saccades.

This achievement would increase its biggest advantage, namely that all the redundant images of a gaze-controlled eye may be taken while fixating a fast moving object by using both inertial sensor data from perturbations of the vehicle itself (measured by tiny devices on the gaze platform) and from the estimated 3-D motion of the object of special interest.

## 5. Conclusions and Outlook

Based on four decades of experience with real-time dynamic machine vision systems, two types of 'vehicle eyes' have been sketched here. In the last two decades of last century, seven test vehicles, two of our institute [54] and five of our partners in industry have been equipped with the latest vision- and computer hardware. Taking the developments in cameras and μP in this century into account, the present state allows completely new types of high-performance-eyes for vehicles: One with all cameras closely collected together and mounted in a fixed way onto the body of the vehicle, and one with a set of five smaller cameras mounted onto a gaze controllable platform. For this gaze-controllable eye, according to Figures 18, 19 and 22, no pixel-computations need to be done at all in order to obtain available tri-focal images of the momentary region of interest. A medium-resolution image is available with a FoV of ±27.5° laterally and ±15.5° vertically around the actual gaze direction. Two low-resolution images from each eye provide multiple redundant image data for the entire frontal hemisphere laterally and ±31° vertically. Each side of the vehicle beyond 90° can be covered entirely by one low-resolution image. The medium-resolution image may cover lateral angles of up to ±117.5°, while high resolution may be available up to ±94.6°; these two angular ranges are of special interest when approaching crossroads.

A FoV of ±25° around the line of sight is covered in a triple-redundant way by each of the eyes (from two low- and one medium-resolution camera), thereby improving safety aspects in the case of a failure of cameras. This compactness, in combination with object-tracking and dynamic scene understanding exploiting the 4-D approach based on spatiotemporal models for motion processes in the real 3-D world, yields an extremely efficient method for real-time vision. It may be the direct basis for some kind of consciousness.

In the other setup, for the body-fix eye, which includes two low-resolution cameras similar to the ones used with gaze control, the increase in data volume for covering the same FoV corresponds to about 0.054%. Evaluating this relatively very small amount of data for features indicating potential objects of interest may considerably reduce the needed access to regions for high-resolution images. For a single object tracked at high resolution in the corresponding image, only about 0.1% of computing operations is needed as compared to the pyramid approach. Thus, the additional low-resolution images and their coarse evaluation for indications of potential objects of interest seem favorable. A complication for cameras mounted in a fixed way onto the body may be the transition when objects move from one image into that of another high-resolution camera. However, from a data volume and data rate point of view, the type of body-fixed eye including redundant low-resolution covering may, as discussed, be competitive with the gaze-controlled eye if motion blur should turn out not to be a challenge in machine vision (in contrast to biological vision systems).

Since this can only be judged properly by comparing actually built devices, this question has to be left open for now. The disadvantages of gaze-controlled vision systems are twofold: first, the mechanisms for precisely pointing the eye in pan and tilt (yaw and pitch); and second, the need for the transfer of the image data from the sensor elements to the vehicle body. If both challenges can be solved by non-mechanical links in the future (say magnetic torques and electromagnetic waves between the eye and its mounting in the vehicle body), this would favor gaze control.

The expectation is that the development of a variety of eyes for vehicles will take up the entire 21st century, similar to the development of the different types of ground vehicles in the 20th century. The requirements depend heavily on the type of mission to be performed. For planetary exploration, a reliable, multi-focal eye with its redundancy will have advantages. On Earth, this may also be true for missions after some catastrophe.

**Funding:** This research received no external funding.

**Conflicts of Interest:** The author declares no conflict of interest

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
