# Peer review of "May a Pair of ‘Eyes’ Be Optimal for Vehicles Too?"

_electronics, doi:10.3390/electronics9050759_

Round 1

Reviewer 1 Report

The title of the article is really good to convince potential readers to go through it. Moreover, the author is a very well-know and reputed professor, in fact, one of the main pioneers, on the use of computer vision for driver assistance and self-driving.

After reading the paper, we can see it consists of a mixture of historic facts of the field and the topic pointed out in the title. At the end of the day, the answer to the question is a sort of “it depends”. The paper includes detailed descriptions on how a self-driving vehicle would be compared to another based on an active vision setting. However, due to the gigantic difficulty that this would involve, there are not real experiments from which deriving conclusions.

Nowadays this kind of work could be presented in the form of a blog, book chapter, or an arxiv report, since it is not bringing advances on any specific topic. But on the other hand, it is fair to acknowledge the enormous work behind the words, the questions, and the explanations highlighted in this paper. Therefore, given the open nature of the Electronics journal, I think this work can be published as a paper.

Said that, I would like some comments to be taken into account:

A. Main Points:

1) There is a detailed description of how a vehicle based on fixed cameras would be with respect to one based on active vision. However, as it is, if a reader would like to do some experimental work (e.g., using a simulator) for comparing both approaches, still it is not straightforward to understand which would be the exact vision systems that the author suggests to compare. It would be valuable if, assuming the use of a simulator (no matter which, just talking in generic terms) the author could describe the vision systems to be compared. Like “Static setting: place a camera (defined by resolution, FoV, HoV, etc) in position/orientation P/O, another in … ; Active setting: place a camera ...”. Somehow, this can be derived from the current explanations, but it would be great for readers to see a sort of table summarizing two systems that would be fair and interesting to compare according to the large experience of the author.

2) Nowadays, most common LiDARs in self-driving prototypes rely on rotating mechanical elements. The industry does not like this due to maintenance issues, and tries to come up with solid state LiDARs for removing mechanical dependencies. Active vision requires moving the cameras, obviously, far less than what is done for LiDARs, but still this may be seen as problematic. I think the author should include a brief discussion about this. A bit is commented on Lines 818-828. However, it would be interesting to contrast the mechanical requirements of an active vision system vs a usual 360º LiDAR.

B. Minor Points:

1) In the 1st paragraph of the introduction, it is said “Neural net approaches are not taken into account for various reasons.” On the one hand, such reasons are not clear and, on the other hand, the paper is not focusing on the understanding of image content. It looks like the author tries to justify why it is not included any mention to what everybody is using nowadays in the field, but I find it unnecessary. Therefore, I would just remove this sentence about neural networks.

3) What means “A-frame”? readers have to figure out themselves from the use of the word here and there, it is better to define it from the first time it appears.

3) In line 226 it is said “At the same conference …”, what conference?

4) In several places “-“ is used but not clear why: Line 286 “Inter- face”; Line 316: -avoidance; Line 444: -recognition and -tracking; Line 464: “..Loop- (HIL-).

Author Response

Main Point 1: You asked for two specific vision systems to be compared: I would start the comparison exactly with the two systems described in Figure 13 (mounted fix) and in Figure 15 with Table 1; all essential parameters are given there. The positioning of both eyes is described in Figure 20.

Main Point2: Both, the types of motion and the goal of their use are completely different for LiDARs and active eyes. The motion of the former one is simple, just a fix rotation rate around a vertical axis. If the unit is mounted fix onto the body, this of course is steadily perturbed by the rotatory motion of the vehicle. All measurement points have to be corrected to eliminate these effects. This becomes more and more difficult with increasing range and amplitude of perturbations. Thus, the high lateral resolution of a tele-camera will never be practically possible by LiDAR. Our human vision system, and a technical one equivalent with 0.2 mrad visual acuity, will not work under strong perturbations without inertial stabilization and object tracking. Therefore, if human-like performance is expected from a technical vision system, there is no way around active gaze control. For many less demanding tasks active vision may not be necessary.

I do not have located this comment at the end of the paper, since I am not convinced that it should be included at all. A thorough discussion would take too much space.

Minor Point 1: The sentence has been removed.

Minor Point 2: Your suggestion has been followed

Minor Point 3: Has been clarified.

Minor Points 4: “-” has been used as repetition sign; at some places I have inserted the original term for a second time.

Reviewer 2 Report

First of all, it is revealed that this paper is a survey paper, not a technical paper. The author analyzed and introduced the sensor system attempted to make an intelligent vehicle (autonomous vehicle) from the past to the present. Unfortunately, the conclusions of this paper do not provide any insights that may be helpful at this point, or the new direction is not written prominently.

// Minor points
1) All pictures should be changed to increase text and increase readability.
2) References should be formatted.

Author Response

Quote: “Unfortunately, the conclusions of this paper do not provide any insights that may be helpful at this point, or the new direction is not written prominently.”

My intention was to make two concrete proposals for the arrangement of cameras in road vehicles, together yielding a pair of “eyes” to be positioned outside the upper left and right end of the front windshield. The design is sketched in Figures 13 to 22, with Figures 16 and 17 documenting experimental results with our test vehicles VaMoRs and VaMP in the 1990s. {I really don’t understand your comment.}

Minor point 1) has been tested on my computer; all images may be expanded by drawing the corners diagonally. I have provided a copy of my personal version to Mr. Yan.

Minor point 2) has been performed.

Reviewer 3 Report

This is a very interesting position paper on the use of bioinspired vision system for autonomous driver (AD). A comprehensive overview of the history of AD is provided giving very good background for the discussion. The differences between the contemporary sensor systems and the foveated vision of vertebrates are analyzed. Two well grounded propositions for vision based systems using either fixed cameras or mobile ones are given. The paper is well written, easy to follow and paints a picture of potential future developments in the area of vision based autonomous driving

Author Response

Thank you very much!